# FlashAttention-3: Fast and Accurate Attention with Asynchrony and Low-precision

**Jay Shah**[*1], **Ganesh Bikshandi**[*1], **Ying Zhang** [2], **Vijay Thakkar** [3,4], **Pradeep Ramani** [3], **Tri Dao**[5,6]

[1] Colfax Research, [2] Meta, [3] NVIDIA, [4] Georgia Institute of Technology
[5] Princeton University, [6] Together AI
{jayhshah,ganesh}@colfax-intl.com, yingz@meta.com,
{vithakkar,prraman}@nvidia.com, tri@tridao.me

## Abstract

Attention, as a core layer of the ubiquitous Transformer architecture, is the bottleneck for large language models and long-context applications. FLASHATTENTION elaborated an approach to speed up attention on GPUs through minimizing memory reads/writes. However, it has yet to take advantage of new capabilities present in recent hardware, with FLASHATTENTION-2 achieving only 35% utilization on the H100 GPU. We develop three main techniques to speed up attention on Hopper GPUs: exploiting asynchrony of the Tensor Cores and TMA to (1) overlap overall computation and data movement via warp-specialization and (2) interleave block-wise matmul and softmax operations, and (3) block quantization and incoherent processing that leverages hardware support for FP8 low-precision. We demonstrate that our method, FLASHATTENTION-3, achieves speedup on H100 GPUs by 1.5-2.0× with BF16 reaching up to 840 TFLOPs/s (85% utilization), and with FP8 reaching 1.3 PFLOPs/s. We validate that FP8 FLASHATTENTION-3 achieves 2.6× lower numerical error than a baseline FP8 attention.

## 1 Introduction

For the Transformer architecture [58], the attention mechanism constitutes the primary computational bottleneck, since computing the self-attention scores of queries and keys has quadratic scaling in the sequence length. Scaling attention to longer context will unlock new capabilities (modeling and reasoning over multiple long documents [24, 43, 49] and files in large codebases [30, 47]), new modalities (high-resolution images [10], audio [23], video [25]), and new applications (user interaction with long history [52], agent workflow with long horizon [61]). This has generated significant interest in making attention faster in the long-context regime, including by approximation [13, 27, 55] and software optimization ([16, 29, 45]), or even alternative architectures [22, 42, 54].

In this work, we build on the work of Dao et al. [16] on developing exact-attention algorithms that integrate knowledge of the GPU's execution model and hardware characteristics into their high-level design. In [16], Dao et al. introduced FLASHATTENTION, a novel tiling strategy for parallelizing attention that eliminates intermediate reads/writes to slow global memory through fusing all of the attention operations into a single GPU kernel. Dao [14] restructured the algorithm as FLASHATTENTION-2 to also parallelize over the sequence length dimension and perform the inner loop of the forward pass over blocks of the key and value matrices, thus improving the occupancy and distribution of work on the GPU. However, we observe that FLASHATTENTION-2 nonetheless achieves poor utilization on newer GPUs relative to optimized matrix-multiplication (GEMM) kernels, such as 35% vs. 80-85% on the Hopper H100 GPU. Partially, this may be attributed to implementation-level differences, such as not using Hopper-specific instructions in place of Ampere ones when targeting the Tensor Cores. Several

---

[*]Equal contribution

38th Conference on Neural Information Processing Systems (NeurIPS 2024).

work such as ThunkerKitten [51] and cuDNN 9 [39] has shown that with Hopper-specific instructions and tile-based abstractions, one can speedup attention computation and simplify the implementation.

More fundamentally, FLASHATTENTION-2's algorithm adheres to a simplified synchronous model and makes no explicit use of asynchrony and low-precision in its design. Asynchrony is a result of hardware specialization to accelerate the most important operations in a ML workload: specific hardware units performing matrix multiplication (Tensor Cores) or memory loading (Tensor Memory Accelerator – TMA), separate from the rest of the CUDA cores performing logic, integer, and floating point computation. Low precision such as FP8 in Hopper and FP4 in Blackwell, continuing the trend of FP16 (Pascal in 2017) and BF16 (Ampere in 2020), is a proven technique to get double or quadruple throughput for the same power and chip area. We review the capabilities afforded by Hopper in these directions in §2.2. The technical challenge is to redesign FLASHATTENTION-2 to make use of these hardware features: asynchrony requires overlapping computation between matmul and softmax even though one depends on the output of the other, and low-precision requires care to minimize quantization error, especially in the case of outlier features in LLMs [20, 53].

To this end, we propose FLASHATTENTION-3, which contributes and synthesizes three new ideas to further improve performance on newer GPU architectures:[2]

1. **Producer-Consumer asynchrony:** We define a warp-specialized software pipelining scheme that exploits the asynchronous execution of data movement and Tensor Cores by splitting producers and consumers of data into separate warps, thereby extending the algorithm's ability to hide memory and instruction issue latencies.
2. **Hiding softmax under asynchronous block-wise GEMMs:** We overlap the comparatively low-throughput non-GEMM operations involved in softmax, such as floating point multiply-add and exponential, with the asynchronous WGMMA instructions for GEMM. As part of this, we rework the FLASHATTENTION-2 algorithm to circumvent certain sequential dependencies between softmax and the GEMMs. For example, in the 2-stage version of our algorithm, while softmax executes on one block of the scores matrix, WGMMA executes in the asynchronous proxy to compute the next block.
3. **Hardware-accelerated low-precision GEMM:** We adapt the forward pass algorithm to allow for targeting the FP8 Tensor Cores for GEMM, nearly doubling the measured TFLOPs/s. This requires bridging the different layout conformance requirements of WGMMA in terms of how blocks of FP32 accumulator and FP8 operand matrices are assumed to be laid out in memory. We use the techniques of block quantization and incoherent processing to mitigate the loss of accuracy that results from moving to FP8 precision.

To validate our method empirically, we benchmark FLASHATTENTION-3 on the H100 SXM5 GPU over a range of parameters and show that (1) BF16 achieves 1.5-2.0× speedup over FLASHATTENTION-2 in the forward pass (reaching up to 840 TFLOPs/s) and 1.5-1.75× in the backward pass, (2) FP8 achieves 1.3 PFLOPs/s, and (3) for large sequence length, BF16 outperforms and FP8 is on par compared to the state-of-the-art implementation of attention from NVIDIA's cuDNN library. We also validate that FP16 FLASHATTENTION-3 yields the same numerical error as FLASHATTENTION-2 and is better than the standard attention implementation as intermediate results (e.g., softmax rescaling) are kept in FP32. Moreover, FP8 FLASHATTENTION-3 with block quantization and incoherent processing is 2.6× more accurate than standard attention with per-tensor quantization in cases with outlier features.

We open-source FLASHATTENTION-3 with a permissive license[3] and plan to integrate it with PyTorch to benefit the largest number of researchers and developers.

## 2 Background: Multi-Head Attention and GPU Characteristics

### 2.1 Multi-Head Attention

Let $\mathbf{Q}, \mathbf{K}, \mathbf{V} \in \mathbb{R}^{N \times d}$ be the query, key and value input sequences associated to a single head, where $N$ is the sequence length and $d$ is the head dimension. Then the attention output $\mathbf{O}$ is computed as:

$$\mathbf{S} = \alpha \mathbf{Q} \mathbf{K}^\top \in \mathbb{R}^{N \times N}, \quad \mathbf{P} = \text{softmax}(\mathbf{S}) \in \mathbb{R}^{N \times N}, \quad \mathbf{O} = \mathbf{P} \mathbf{V} \in \mathbb{R}^{N \times d},$$

where softmax is applied row-wise and one typically sets $\alpha = 1/\sqrt{d}$ as the scaling factor. In practice, we subtract rowmax($\mathbf{S}$) from $\mathbf{S}$ to prevent numerical instability with the exponential function. For

---

[2]We describe our results in the context of NVIDIA's Hopper architecture. However, our algorithm is operative for any GPU architecture with sufficiently robust asynchronous execution and low-precision capabilities.

[3]FLASHATTENTION-3 is available at `https://github.com/Dao-AILab/flash-attention`

multi-head attention (MHA), each head has its own set of query, key and value projections, and this computation parallelizes across multiple heads and batches to produce the full output tensor.

Now let $\phi$ be a scalar loss function and let $\mathbf{d}(-) = \partial\phi/\partial(-)$ be notation for the gradient. Given the output gradient $\mathbf{dO} \in \mathbb{R}^{N \times d}$, we compute $\mathbf{dQ}$, $\mathbf{dK}$, and $\mathbf{dV}$ according to the chain rule as follows:

$$\mathbf{dV} = \mathbf{P}^\top \mathbf{dO} \in \mathbb{R}^{N \times d}, \qquad\qquad \mathbf{dP} = \mathbf{dOV}^\top \in \mathbb{R}^{N \times N},$$

$$\mathbf{dS} = \mathrm{dsoftmax}(\mathbf{dP}) \in \mathbb{R}^{N \times N}, \qquad \mathbf{dQ} = \alpha\mathbf{dSK} \in \mathbb{R}^{N \times d}, \qquad \mathbf{dK} = \alpha\mathbf{dS}^\top\mathbf{Q} \in \mathbb{R}^{N \times d}.$$

Here, we have that $\mathbf{d}s = (\mathrm{diag}(p) - pp^\top)\mathbf{d}p$ for $p = \mathrm{softmax}(s)$ as a function of a vector $s$, and we write $\mathrm{dsoftmax}(\mathbf{dP})$ for this formula applied row-wise. Finally, this computation again parallelizes across the number of heads and batches for the backward pass of MHA.

## 2.2 GPU hardware characteristics and execution model

We describe the aspects of the GPU's execution model relevant for FLASHATTENTION-3, with a focus on the NVIDIA Hopper architecture as a concrete instantiation of this model.

**Memory hierarchy:** The GPU's memories are organized as a hierarchy of data locales, with capacity inversely related to bandwidth (Table 1)[4]. Global memory (GMEM), also known as HBM, is the off-chip DRAM accessible to all streaming multiprocessors (SMs). Data from GMEM gets transparently cached into an on-chip L2 cache. Next, each SM contains a small on-chip, programmer-managed highly banked cache called shared memory (SMEM). Lastly, there is the register file within each SM.

**Thread hierarchy:** The GPU's programming model is organized around logical groupings of execution units called threads. From the finest to coarsest level, the thread hierarchy is comprised of threads, warps (32 threads), warpgroups (4 contiguous warps), threadblocks (i.e., cooperative thread arrays or CTAs), threadblock clusters (in Hopper), and grids.

These two hierarchies are closely interlinked. Threads in the same CTA are co-scheduled on the same SM, and CTAs in the same cluster are co-scheduled on the same GPC. SMEM is directly addressable by all threads within a CTA, whereas each thread has at most 256 registers (RMEM) private to itself.

Table 1: Thread-Memory hierarchy for the NVIDIA Hopper H100 SXM5 GPU.

| Hardware Level | Parallel Agent | Data Locale | Capacity @ Bandwidth |
|---:|---|---|---|
| Chip | Grid | GMEM | 80 GiB @ 3.35 TB/s |
| GPC | Threadblock Clusters | L2 | 50 MiB @ 12 TB/s |
| SM | Threadblock (CTA) | SMEM | 228 KiB per SM, 31TB/s per GPU |
| Thread | Thread | RMEM | 256 KiB per SM |

**Asynchrony and warp-specialization:** GPUs are throughput processors that rely on concurrency and asynchrony to hide memory and execution latencies. For async memory copy between GMEM and SMEM, Hopper has the Tensor Memory Accelerator (TMA) as a dedicated hardware unit [38, §7.29]. Furthermore, unlike prior architectures such as Ampere, the Tensor Core of Hopper, exposed via the warpgroup-wide WGMMA instruction [40, §9.7.14], is also asynchronous and can source its inputs directly from shared memory.

Hardware support for asynchrony allows for warp-specialized kernels, where the warps of a CTA are divided into producer or consumer roles that only ever issue either data movement or computation. Generically, this improves the compiler's ability to generate optimal instruction schedules [4]. In addition, Hopper supports the dynamic reallocation of registers between warpgroups via `setmaxnreg` [40, §9.7.17.1], so those warps doing MMAs can obtain a larger share of RMEM than those just issuing TMA (for which only a single thread is needed).

**Low-precision number formats:** Modern GPUs have specialized hardware units for accelerating low-precision computation. For example, the WGMMA instruction can target the FP8 Tensor Cores on Hopper to deliver 2x the throughput per SM when compared to FP16 or BF16.

---

[4]Luo et al. [34] reports shared memory bandwidth of 128 bytes per clock cycle per SM, and we multiply that by 132 SMs and the boost clock of 1830 MHz.

However, correctly invoking FP8 WGMMA entails understanding the layout constraints on its operands. Given a GEMM call to multiply $A \times B^\top$ for an $M \times K$-matrix $A$ and an $N \times K$-matrix $B$, we say that the $A$ or $B$ operand is *mn-major* if it is contiguous in the outer $M$ or $N$ dimension, and *k-major* if is instead contiguous in the inner $K$-dimension. Then for FP16 WGMMA, both mn-major and k-major input operands are accepted for operands in SMEM, but for FP8 WGMMA, only the k-major format is supported. Moreover, in situations such as attention where one wants to fuse back-to-back GEMMs in a single kernel, clashing FP32 accumulator and FP8 operand layouts pose an obstacle to invoking dependent FP8 WGMMAs.

In the context of attention, these layout restrictions entail certain modifications to the design of an FP8 algorithm, which we describe in §3.3.

## 2.3 Standard Attention and Flash Attention

Following Dao et al. [16], we let **standard attention** denote an implementation of attention on the GPU that materializes the intermediate matrices $\mathbf{S}$ and $\mathbf{P}$ to HBM. The main idea of FLASHATTENTION was to leverage a local version of the softmax reduction to avoid these expensive intermediate reads/writes and fuse attention into a single kernel. Local softmax corresponds to lines 18-19 of the consumer mainloop in Algorithm 1 together with the rescalings of blocks of $\mathbf{O}$. The simple derivation that this procedure indeed computes $\mathbf{O}$ can be found in [14, §2.3.1].

# 3 FlashAttention-3: Algorithm

In this section, we describe the FLASHATTENTION-3 algorithm. For simplicity, we focus on the forward pass, with the backward pass algorithm described in Appendix B.1. We first indicate how to integrate warp-specialization with a circular SMEM buffer into the base algorithm of FLASHATTENTION-2. We then explain how to exploit asynchrony of WGMMA to define an overlapped GEMM-softmax 2-stage pipeline. Finally, we describe the modifications needed for FP8, both in terms of layout conformance and accuracy via block quantization and incoherent processing.

## 3.1 Producer-Consumer asynchrony through warp-specialization and pingpong scheduling

**Warp-specialization** As with FLASHATTENTION-2, the forward pass of FLASHATTENTION-3 is embarrassingly parallel in the batch size, number of heads, and query sequence length. Thus, it will suffice to give a CTA-level view of the algorithm, which operates on a tile $\mathbf{Q}_i$ of the query matrix to compute the corresponding tile $\mathbf{O}_i$ of the output. To simplify the description, we first give the warp-specialization scheme with a circular SMEM buffer that does *not* have in addition the GEMM-softmax overlapping. Let $d$ be the head dimension, $N$ the sequence length, and fix a query block size $B_r$ to divide $\mathbf{Q}$ into $T_r = \lceil \frac{N}{B_r} \rceil$ blocks $\mathbf{Q}_1,..,\mathbf{Q}_{T_r}$.

For our implementation of Algorithm 1 on Hopper, we use `setmaxnreg` for (de)allocations, TMA for loads of $\mathbf{Q}_i$ and $\{\mathbf{K}_j,\mathbf{V}_j\}_{0 \leq j < T_c}$, and WGMMA to execute the GEMMs in the consumer mainloop, with the SS or RS prefix indicating whether the first operand is sourced from shared memory or register file. For interpreting the execution flow of Algorithm 1, note that issuing TMA loads does not stall on the completion of other loads due to asynchrony. Moreover, in the producer mainloop, no waits will be issued for the first $s$ iterations as the buffer gets filled.

**Pingpong scheduling** The asynchronous nature of WGMMA and TMA, along with warp-specialization, opens up the opportunity to overlap the softmax computation of one warpgroup with the GEMM of another warpgroup. To motivate this, notice that non-matmul operations have much lower throughput than matmul operations on modern hardware accelerators. As an example, the H100 SXM5 GPU has 989 TFLOPS of FP16 matmul but only 3.9 TFLOPS of special functions such as exponential[5] (necessary for softmax). For the attention forward pass in FP16 with head dimension 128, there are 512x more matmul FLOPS compared to exponential operations, but the exponential has 256x lower throughput, so exponential can take 50% of the cycle compared to matmul. The situation is even worse with FP8, where the matmul throughput doubles but the exponential throughput stays the same.

---

[5]The CUDA programming guide specifies that 16 operations of special functions can be performed per streaming multiprocessor (SM) per clock cycle. We multiply 16 by 132 SMs and 1830 MHz clock speed to get 3.9 TFLOPS of special functions.

**Algorithm 1** FLASHATTENTION-3 forward pass **without** intra-consumer overlapping – CTA view

**Require:** Matrices $\mathbf{Q}_i \in \mathbb{R}^{B_r \times d}$ and $\mathbf{K}, \mathbf{V} \in \mathbb{R}^{N \times d}$ in HBM, key block size $B_c$ with $T_c = \lceil \frac{N}{B_c} \rceil$.

 1: Initialize pipeline object to manage barrier synchronization with $s$-stage circular SMEM buffer.
 2: **if** in producer warpgroup **then**
 3:     Deallocate predetermined number of registers.
 4:     Issue load $\mathbf{Q}_i$ from HBM to shared memory.
 5:     Upon completion, commit to notify consumer of the load of $\mathbf{Q}_i$.
 6:     **for** $0 \le j < T_c$ **do**
 7:         Wait for the $(j\%s)$th stage of the buffer to be consumed.
 8:         Issue loads of $\mathbf{K}_j, \mathbf{V}_j$ from HBM to shared memory at the $(j\%s)$th stage of the buffer.
 9:         Upon completion, commit to notify consumers of the loads of $\mathbf{K}_j, \mathbf{V}_j$.
10:     **end for**
11: **else**
12:     Reallocate predetermined number of registers as function of number of consumer warps.
13:     On-chip, initialize $\mathbf{O}_i = (0) \in \mathbb{R}^{B_r \times d}$ and $\ell_i, m_i = (0), (-\infty) \in \mathbb{R}^{B_r}$.
14:     Wait for $\mathbf{Q}_i$ to be loaded in shared memory.
15:     **for** $0 \le j < T_c$ **do**
16:         Wait for $\mathbf{K}_j$ to be loaded in shared memory.
17:         Compute $\mathbf{S}_i^{(j)} = \mathbf{Q}_i \mathbf{K}_j^T$ (SS-GEMM). Commit and wait.
18:         Store $m_i^{\text{old}} = m_i$ and compute $m_i = \max(m_i^{\text{old}}, \text{rowmax}(\mathbf{S}_i^{(j)}))$.
19:         Compute $\widetilde{\mathbf{P}}_i^{(j)} = \exp(\mathbf{S}_i^{(j)} - m_i)$ and $\ell_i = \exp(m_i^{\text{old}} - m_i)\ell_i + \text{rowsum}(\widetilde{\mathbf{P}}_i^{(j)})$.
20:         Wait for $\mathbf{V}_j$ to be loaded in shared memory.
21:         Compute $\mathbf{O}_i = \text{diag}(\exp(m_i^{\text{old}} - m_i))\mathbf{O}_i + \widetilde{\mathbf{P}}_i^{(j)}\mathbf{V}_j$ (RS-GEMM). Commit and wait.
22:         Release the $(j\%s)$th stage of the buffer for the producer.
23:     **end for**
24:     Compute $\mathbf{O}_i = \text{diag}(\ell_i)^{-1}\mathbf{O}_i$ and $L_i = m_i + \log(\ell_i)$.
25:     Write $\mathbf{O}_i$ and $L_i$ to HBM as the $i$th block of $\mathbf{O}$ and $L$.
26: **end if**

Since the exponential is performed by a separate hardware unit (the multi-function unit), ideally we'd want the exponential calculation to be scheduled when the Tensor Cores are performing the matmul. To do so, we use synchronization barriers (`bar.sync` instructions) to force the GEMMs (GEMM1 – $\mathbf{PV}$ of one iteration, and GEMM0 – $\mathbf{QK}^\top$ of the next iteration) of warpgroup 1 to be scheduled before the GEMMs of warpgroup 2. As a result, the softmax of warpgroup 1 will be scheduled while warpgroup 2 is performing its GEMMs. Then the roles swap, with warpgroup 2 doing softmax while warpgroup 1 doing GEMMs (hence, "pingpong" scheduling). This is illustrated in Fig. 1. Though in practice the pingpong scheduling is not as clean as depicted in the figure, we generally find this to improve performance (e.g., from 570 TFLOPS to 620-640 TFLOPS for FP16 forward with head dimension 128 and sequence length 8192).

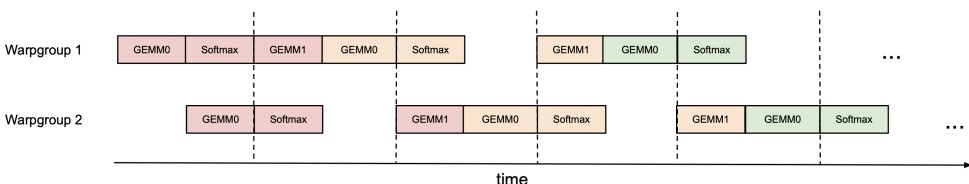

Figure 1: Pingpong scheduling for 2 warpgroups to overlap softmax and GEMMs: the softmax of one warpgroup should be scheduled when the GEMMs of another warpgroup are running. The same color denotes the same iteration.

**Attention variants** For multi-query attention [50] and grouped query attention [3], we follow the approach in FLASHATTENTION-2 and adjust the tensor indexing to avoid duplicating $\mathbf{K}$ and $\mathbf{V}$ in HBM.

## 3.2 Intra-warpgroup overlapping GEMMs and softmax

Even within one warpgroup, we can overlap some instructions in the softmax with some instructions in the GEMMs. We describe one technique to do so.

In the attention algorithm, operations within the inner loop (main loop) have sequential dependencies that impede parallelization within a single iteration. For example, (local) softmax (lines 18 to 19) relies on the output $\mathbf{S}_i^{(j)}$ of the first GEMM, while the second GEMM takes its result $\widetilde{\mathbf{P}}_i^{(j)}$ as an operand. Indeed, the wait statements in lines 17 and 21 of Algorithm 1 serialize the execution of softmax and GEMMs. However, we can break these dependencies by pipelining across iterations through additional buffers in registers. Pursuing this idea, we propose the following two-stage[6] GEMM-softmax pipelining algorithm:

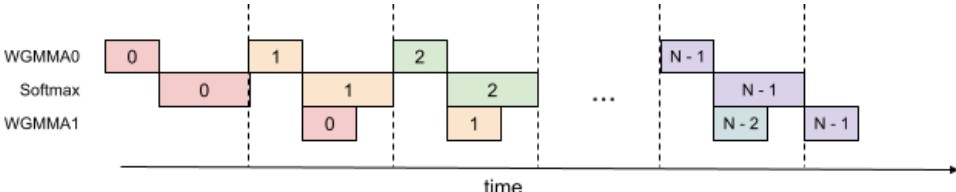

Figure 2: 2-stage WGMMA-softmax pipelining

---

**Algorithm 2** FLASHATTENTION-3 consumer warpgroup forward pass

---

**Require:** Matrices $\mathbf{Q}_i \in \mathbb{R}^{B_r \times d}$ and $\mathbf{K}, \mathbf{V} \in \mathbb{R}^{N \times d}$ in HBM, key block size $B_c$ with $T_c = \lceil \frac{N}{B_c} \rceil$.
 1: Reallocate predetermined number of registers as function of number of consumer warps.
 2: On-chip, initialize $\mathbf{O}_i = (0) \in \mathbb{R}^{B_r \times d}$ and $\ell_i, m_i = (0), (-\infty) \in \mathbb{R}^{B_r}$.
 3: Wait for $\mathbf{Q}_i$ and $\mathbf{K}_0$ to be loaded in shared memory.
 4: Compute $\mathbf{S}_{\text{cur}} = \mathbf{Q}_i \mathbf{K}_0^T$ using WGMMA. Commit and wait.
 5: Release the 0th stage of the buffer for $\mathbf{K}$.
 6: Compute $m_i$, $\tilde{\mathbf{P}}_{\text{cur}}$ and $\ell_i$ based on $\mathbf{S}_{\text{cur}}$, and rescale $\mathbf{O}_i$.
 7: **for** $1 \leq j < T_c - 1$ **do**
 8:     Wait for $\mathbf{K}_j$ to be loaded in shared memory.
 9:     Compute $\mathbf{S}_{\text{next}} = \mathbf{Q}_i \mathbf{K}_j^T$ using WGMMA. Commit but do not wait.
10:     Wait for $\mathbf{V}_{j-1}$ to be loaded in shared memory.
11:     Compute $\mathbf{O}_i = \mathbf{O}_i + \tilde{\mathbf{P}}_{\text{cur}} \mathbf{V}_{j-1}$ using WGMMA. Commit but do not wait.
12:     Wait for the WGMMA $\mathbf{Q}_i \mathbf{K}_j^T$.
13:     Compute $m_i$, $\tilde{\mathbf{P}}_{\text{next}}$ and $\ell_i$ based on $\mathbf{S}_{\text{next}}$.
14:     Wait for the WGMMA $\tilde{\mathbf{P}}_{\text{cur}} \mathbf{V}_{j-1}$ and then rescale $\mathbf{O}_i$
15:     Release the $(j \% s)$th, resp. $(j-1 \% s)$th stage of the buffer for $\mathbf{K}$, resp. $\mathbf{V}$.
16:     Copy $\mathbf{S}_{\text{next}}$ to $\mathbf{S}_{\text{cur}}$.
17: **end for**
18: Wait for $\mathbf{V}_{T_c-1}$ to be loaded in shared memory.
19: Compute $\mathbf{O}_i = \mathbf{O}_i + \tilde{\mathbf{P}}_{\text{last}} \mathbf{V}_{T_c-1}$ using WGMMA. Commit and wait.
20: Epilogue: Rescale $\mathbf{O}_i$ based on $m_i$. Compute $L_i$ based on $m_i$ and $\ell_i$. Write $\mathbf{O}_i$ and $L_i$ to HBM as the $i$-th block of $\mathbf{O}$ and $L$.

---

Algorithm 2 functions as a replacement for the consumer path of Algorithm 1 to comprise the complete FLASHATTENTION-3 algorithm for FP16 precision. At a high-level, we use WGMMA as a metonym for asynchronous GEMM. Within the mainloop (lines 8 to 16), the second WGMMA operation of iteration $j$ (line 11) is overlapped with softmax operations from iteration $j+1$ (line 13).

While the pipelined structure illustrated above offers theoretical performance gains, there are several practical aspects to consider:

---

[6]Note that the number of stages of the overlapping scheme is bounded by, but need not equal, the number $s$ of stages in the circular SMEM buffer.

| T0 {d0, d1} | T1 {d0, d1} | T2 {d0, d1} | T3 {d0, d1} | T0 {d4, d5} | T1 {d4, d5} | T2 {d4, d5} | T3 {d4, d5} |
|---|---|---|---|---|---|---|---|
| T0 {d2, d3} | T1 {d2, d3} | T2 {d2, d3} | T3 {d2, d3} | T0 {d6, d7} | T1 {d6, d7} | T2 {d6, d7} | T3 {d6, d7} |

Figure 3: FP32 accumulator register WGMMA layout – rows 0 and 8, threads 0-3, entries 0-7.

| T0 {a0, a1} | T0 {a2, a3} | T1 {a0, a1} | T1 {a2, a3} | T2 {a0, a1} | T2 {a2, a3} | T3 {a0, a1} | T3 {a2, a3} |
|---|---|---|---|---|---|---|---|
| T0 {a4, a5} | T0 {a6, a7} | T1 {a4, a5} | T1 {a6, a7} | T2 {a4, a5} | T2 {a6, a7} | T3 {a4, a5} | T3 {a6, a7} |

Figure 4: FP8 operand A register WGMMA layout – rows 0 and 8, threads 0-3, entries 0-7.

**Compiler reordering**    The pseudocode represents an idealized execution order but the compiler (NVCC) often rearranges instructions for optimization. This can disrupt the carefully crafted WGMMA and non-WGMMA operation pipelining sequence, potentially leading to unexpected behavior or diminished performance gains. An analysis of the SASS code shows that the compiler generates overlapped code as expected (Section B.2).

**Register pressure**    To maintain optimal performance, register spilling should be minimized. However, the 2-stage pipeline requires additional registers to store intermediate results and maintain context between stages. Specifically, an extra $S_{next}$ must be kept in registers, leading to extra register usage of size $B_r \times B_c \times \text{sizeof}(\text{float})$ per threadblock. This increased register demand may conflict with using larger block sizes (another common optimization), which is also register-hungry. In practice, trade-offs should be made based on profiling results.

**3-stage pipelining**    Extending the 2-stage algorithm described above, we propose a 3-stage variant that would further overlap the second WGMMA with softmax. While this approach offers the potential for even higher Tensor Core utilization, it requires even more registers due to an additional stage in the pipeline, making the trade-off between tile size and pipeline depth more difficult to balance. A detailed description of the 3-stage algorithm and its evaluation results can be found in  Appendix B.3.

### 3.3   Low-precision with FP8

**Efficiency:   FP8 layout for accumulator and operand.**    Computing the forward pass of FLASHATTENTION-3 in FP8 precision poses two additional challenges not encountered for FP16 in terms of layout conformance. The first relates to the datatype conversion of the first WGMMA's FP32 accumulator to the second WGMMA's lower-precision (FP16 or FP8) operand, which was left implicit in Algorithm 1. Specifically, after downcasting to FP8, we need to transform the register ownership pattern from that depicted in Fig. 3 into Fig. 4, repeated per every four threads of the consumer warpgroups.

In Appendix B.7, we give a solution for this in code using shuffle instructions.

Secondly, the k-major constraint on FP8 WGMMA explained in §2.2 entails clashing assumptions on how $\mathbf{Q}$, $\mathbf{K}$, and $\mathbf{V}$ are laid out in global memory, since the TMA load cannot change the contiguous dimension. Namely, $\mathbf{Q}$ and $\mathbf{K}$ should be contiguous in the head dimension, whereas $\mathbf{V}$ should be contiguous in the sequence length dimension. We perform in-kernel transposition of the $\mathbf{V}_j$ tiles in SMEM prior to invoking the second FP8 WGMMA, since in practice, $\mathbf{V}$ is typically assumed to be contiguous in the head dimension. In Appendix B.8, we describe in details how to perform the $\mathbf{V}$ transpose as part of the attention kernel itself.

**Accuracy: block quantization and incoherent processing.**  With FP8 (e4m3) format, one only uses 3 bits to store the mantissa and 4 bits for the exponent. This results in higher numerical error than FP16/BF16. Moreover, large models typically have outlier values [20, 53] that are much larger in magnitude than most other values, making quantization difficult. One typically use per-tensor scaling [37] by keeping one scalar per tensor (e.g., one for $\mathbf{Q}$, for $\mathbf{K}$, and for $\mathbf{V}$). To reduce the numerical error of attention in FP8, we employ two techniques:

1. **Block quantization**: we keep one scalar per block, so that for each of $\mathbf{Q}$, $\mathbf{K}$, $\mathbf{V}$ we split the tensor into blocks of size $B_r \times d$ or $B_c \times d$ and quantize them separately. This quantization can be fused with an operation right before attention (e.g., rotary embedding) with no additional slow down (since rotary embedding is memory-bandwidth bound). As the FLASHATTENTION-3 algorithm naturally operates on blocks, we can scale each block of $\mathbf{S}$ to account for this block quantization at no computation cost.

2. **Incoherent processing**: to even out outliers, we multiply $\mathbf{Q}$ and $\mathbf{K}$ with a random orthogonal matrix $\mathbf{M}$ before quantizing to FP8. Since $\mathbf{M}$ is orthogonal, $\mathbf{MM}^\top = I$ and so $(\mathbf{QM})(\mathbf{KM})^\top = \mathbf{QK}^\top$, i.e., multiplying both $\mathbf{Q}$ and $\mathbf{K}$ with $\mathbf{M}$ does not change the attention output. This serves to "spread out" the outliers since each entry of $\mathbf{QM}$ or $\mathbf{KM}$ is a random sum of entries of $\mathbf{Q}$ or $\mathbf{K}$, thus reducing quantization error. In practice, we follow Chee et al. [8] and Tseng et al. [57] and choose $\mathbf{M}$ to be the product of random diagonal matrices of $\pm 1$ and a Hadamard matrix, which can be multiplied in $O(d \log d)$ instead of $O(d^2)$, and can also be fused with the rotary embedding at no extra computation cost.

We validate that these two techniques reduces numerical error by up to 2.6× in §4.3.

## 4  Empirical Validation

We use the primitives from CUTLASS [56] such as WGMMA and TMA abstractions to implement FLASHATTENTION-3 and evaluate its efficiency and accuracy.

- **Benchmarking attention.** We measure the runtime of FLASHATTENTION-3 across different sequence lengths and compare it to a standard implementation in PyTorch, FLASHATTENTION-2, FLASHATTENTION-2 in Triton (which uses H100-specific instructions), as well as a vendor's implementation of FLASHATTENTION-2 optimized for H100 GPUs from cuDNN. We confirm that FLASHATTENTION-3 is up to 2.0× faster than FLASHATTENTION-2 and 1.5× faster than FLASHATTENTION-2 in Triton. FLASHATTENTION-3 reaches up to 840 TFLOPs/s, 85% of the theoretical maximum TFLOPs/s on H100 GPUs.

- **Ablation study.** We confirm that our algorithmic improvements with warp-specialization and GEMM-softmax pipelining contribute to the speedup of FLASHATTENTION-3.

- **Accuracy of FP8 attention.** We validate that block quantization and incoherent processing reduces the numerical error of FP8 FLASHATTENTION-3 by 2.6×.

### 4.1  Benchmarking Attention

We measure the runtime of different attention methods on an H100 80GB SXM5 GPU for different settings (without / with causal mask, head dimension 64 or 128) for BF16 inputs. We report the results in Fig. 5 and Fig. 6, showing that FLASHATTENTION-3 is around 1.5-2.0× faster than FLASHATTENTION-2 in the forward pass and 1.5-1.75× faster in the backward pass. Compared to a standard attention implementation, FLASHATTENTION-3 can be up to 3-16× faster. For medium and long sequences (1k and above), FLASHATTENTION-3 even surpasses the speed of a vendor's library (cuDNN – closed source) that has been optimized for H100 GPUs.

**Benchmark settings:**  We vary the sequence length as 512, 1k, ..., 16k, and set batch size so that the total number of tokens is 16k. We set the hidden dimension to 2048, and head dimension to be either 64, 128, or 256 (i.e., 32 heads, 16 heads, or 8 heads). To calculate the FLOPs of the forward pass, we use:

$$4 \cdot \text{seqlen}^2 \cdot \text{head dimension} \cdot \text{number of heads}.$$

With causal masking, we divide this number by 2 to account for the fact that approximately only half of the entries are calculated. To get the FLOPs of the backward pass, we multiply the forward pass FLOPs by 2.5 (since there are 2 matmuls in the forward pass and 5 matmuls in the backward pass, due to recomputation).

We also measure the runtime for FP8 for the forward pass under similar settings. We report the results for headdim 256 in  Fig. 7 and give the full results in Appendix C.2.

### 4.2  Ablation Study: 2-Stage Pipelining Experiments

We ablate both the 2-stage WGMMA-softmax pipelining and warp-specialization for non-causal FP16 FLASHATTENTION-3 with fixed parameters $\{\text{batch}, \text{seqlen}, \text{nheads}, \text{hdim}\} = \{4, 8448, 16, 128\}$. The result in Table 2 confirms that our algorithmic improvements (asynchrony with warp-specialization and overlapping between GEMM and softmax) lead to significant speedup, from 570 to 661 TFLOPs.

### 4.3  Numerical Error Validation

As there has been interest in the numerical error [21] of FLASHATTENTION, we compare FLASHATTENTION-2, FLASHATTENTION-3, and a standard implementation of attention against

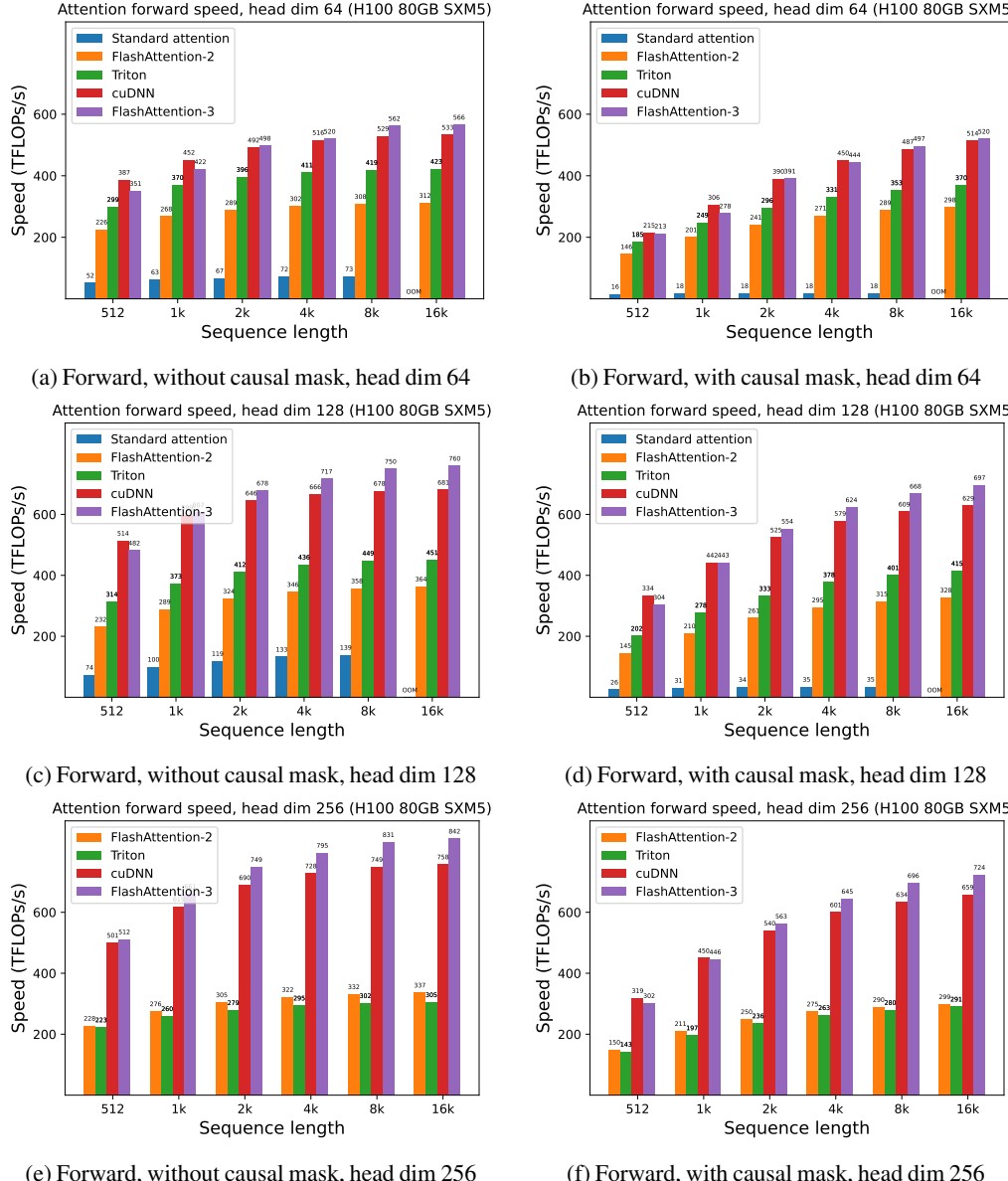

(a) Forward, without causal mask, head dim 64      (b) Forward, with causal mask, head dim 64

(c) Forward, without causal mask, head dim 128      (d) Forward, with causal mask, head dim 128

(e) Forward, without causal mask, head dim 256      (f) Forward, with causal mask, head dim 256

Figure 5: Attention forward speed (BF16) on H100 GPU

Table 2: Pipelining ablation measurements

| Configuration | Time | TFLOPs/s |
|---|---|---|
| FLASHATTENTION-3 | 3.538 ms | 661 |
| No GEMM-Softmax Pipelining, Warp-Specialization | 4.021 ms | 582 |
| GEMM-Softmax Pipelining, No Warp-Specialization | 4.105 ms | 570 |

a reference implementation in FP64. To simulate outlier features and activations in LLMs [20, 53], we generate the entries of **Q**,**K**,**V** with the following distribution:

$$\mathcal{N}(0,1)+\mathcal{N}(0,100)\cdot\text{Bernoulli}(0.001).$$

That is, each entry is normally distributed with zero mean and standard deviation 1, but for 0.1% of entries we add an independent term that's normally distributed with standard deviation 10. We then measure the root mean squared error (RMSE) in Table 3. In FP16, both FLASHATTENTION-2 and FLASHATTENTION-3 achieves 1.7× lower RMSE compared to the standard implementation since intermediate results (softmax) are kept in FP32. The baseline attention in FP8 uses per-tensor scaling, with

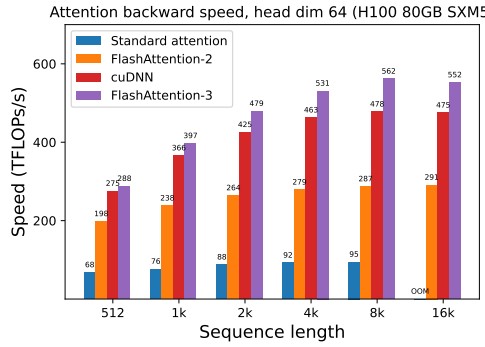
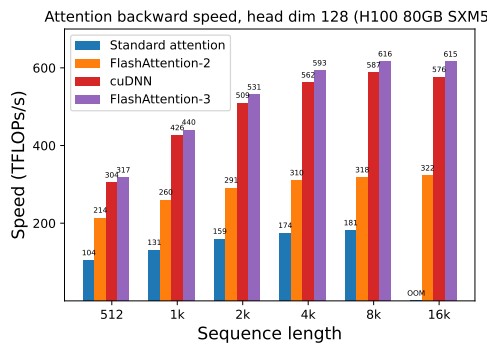

(a) Backward, without causal mask, head dim 64

(b) Backward, without causal mask, head dim 128

Figure 6: Attention backward speed (BF16) on H100 GPU

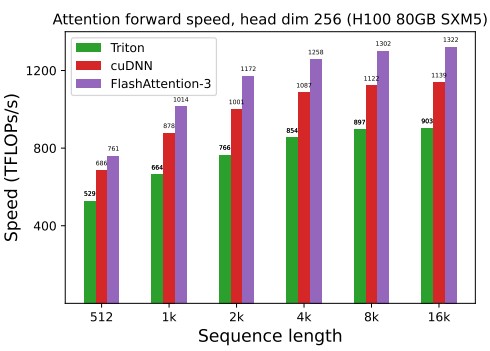
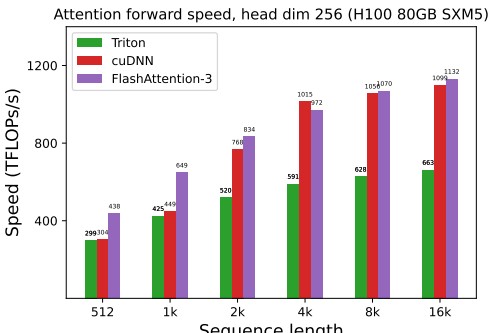

(a) Forward, without causal mask, head dim 256

(b) Forward, with causal mask, head dim 256

Figure 7: Attention forward speed (FP8) on H100 GPU

matmul accumulator in FP32 and intermediate softmax results kept in FP16. Thanks to block quantization and incoherent processing, FLASHATTENTION-3 in FP8 is 2.6× more accurate than this baseline.

Table 3: Numerical error comparisons in FP16 and FP8 (e4m3).

| Method | Baseline FP16 | FLASHATTENTION-2 FP16 | FLASHATTENTION-3 FP16 |
|---|---|---|---|
| RMSE | 3.2e-4 | **1.9e-4** | **1.9e-4** |

| Method | Baseline FP8 | FLASHATTENTION-3 FP8 | No block quant | No incoherent processing |
|---|---|---|---|---|
| RMSE | 2.4e-2 | **9.1e-3** | 9.3e-3 | 2.4e-2 |

## 5 Dicussion, Limitations, Conclusion

With FLASHATTENTION-3, we have demonstrated that new programming techniques and hardware features such as asynchrony and low-precision can have a dramatic impact on the efficiency and accuracy of attention. We are able to speed up attention by 1.5-2.0× times compared to FLASHATTENTION-2, and reduce FP8 numerical error by 2.6× compared to standard per-tensor quantization. Some limitations of our work that we hope to address in the future include: optimizing for LLM inference, and understanding the effects of low-precision attention in large-scale training. Though we have focused on Hopper GPUs in this work, we expect that the techniques developed here will apply to other hardware accelerators. We hope that a faster and more accurate primitive such as attention will unlock new applications in long-context tasks.

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

# A    Related Work

**Attention variants and distributed attention**    Ever since attention became popular with the Transformer architecture [58], there has been a large body of work on approximating attention to scale it to longer sequences. These approximation methods can generally be categorized into two classes: sparse and low-rank. Sparse attention only computes some entries of the attention matrix (softmax($\mathbf{QK}^T$)) and assumes that other entries are zero. Different methods have different ways of choosing which entries should be zero, either with a fixed pattern [11], with a sliding window [6], or with a dynamic pattern through hashing [28] or routing [46]. The low-rank approach instead assumes that the attention matrix has a low-rank structure, and apply a pointwise nonlinearity to the query and key [27] with random projection [12, 44, 60]. One can also combine the sparse and low-rank approximation for better quality [9, 62]. However, these approximation methods typically do not offer the same model quality as standard attention [55], and so most large-scale models do not employ these techniques.

There are other variants of attention aimed at reducing the size of the KV cache to improve inference efficiency. Multi-query attention [50] and grouped query attention [3] tie different heads of $\mathbf{K}$ and $\mathbf{V}$, and multiple query heads interact with the same key and value head. Multi-head latent attention [19] parameterizes the $\mathbf{K}$ and $\mathbf{V}$ as low-rank projections of a shared matrix to further reduce the KV cache size. However, all of these approaches do not change the core computation softmax($\mathbf{QK}^T$)$\mathbf{V}$ during training and simply change how $\mathbf{Q},\mathbf{K},\mathbf{V}$ are obtained. As a result, any efficiency or accuracy improvement to the standard attention computation benefits these methods.

To extend to even longer context, attention computation can be distributed across multiple GPUs. Methods such as Ring attention [31, 32] and variants [7] can reach a context length of up to 1 million. They use FLASHATTENTION (or FLASHATTENTION-2) as a primitive, and so the improvement from FLASHATTENTION-3 would benefit these distributed attention methods as well.

**Alternative architectures**    Motivated by the limitations of attention, a variety of alternative architectures have been proposed. They build on the connection between linear attention [27] and recurrent neural networks (RNNs). RWKV [42], H3 [17], MEGA [35], Retnet [54] enhance the expressivity of the simple cumulative sum in linear attention with more sophisticated recurrences. Mamba [22] and xLSTM [5] use learnable weighting for the recurrence and can match the quality of Transformers in language modeling at small or medium scale. These approaches can be connected to generalizations of linear attention through the lens of the structure of the token-mixing matrix [15]. These models have started to see some traction, seeing usage in some medium to large-scale models such as Jamba [2], Zamba [63], Megalodon [36], and Mamba2-hybrid [59]. For the highest quality, these SSM- and RNN-based models still employ many layers of attention. We expect that techniques to speed up attention presented in this work will be useful to speedup these alternative architectures.

**Low-precision attention**    Quantization is a promising approach to speed up attention, but they have mostly focused on reducing the space for KV cache for inference efficiency. QuIP [8] and QuIP#[57] use incoherent processing to reduce the quantization, and we adapted this technique for FP8 FLASHATTENTION-3. Recent work suggests that for inference the KV cache is highly compressible down to 4-, 3-, or even 2-bits [26, 33]. However, quantization during training is still challenging as higher precision is typically required for stable training.

**Hardware-aware Algorithms**    Our work presented in this paper focuses on the micro-architecture specific tuning to leverage new instruction sets and adopt a natively asynchronous programming model. There are other orthogonal axes for hardware-aware algorithm co-design being explored. A recent example of this is LeanAttention [48], which recognizes the poor GPU occupancy and high memory bandwidth requirements of the sequential token generation phase as primary bottlenecks for inference and optimizes it via a smarter load balancing strategy similar to Stream-K load balancing [41] to achieve nearly peak occupancy. There is a large literature on optimizing GEMM for specific hardware that employs many of the same techniques. As an example, Abdelfattah et al. [1] presents a high performance batched GEMM kernel on K40c Graphics Processing Units (GPU) for both fixed and variable sizes, proposing specialized GEMM designs and a comprehensive autotuning process to deliver state-of-the-art performance.

# B Addition Details on Algorithms

## B.1 Asynchrony Through Warp Specialization for the Backward Pass

Similar to the forward pass §3.1, we use warp specialization to handle asynchrony. Instead of just a simple producer-consumer pattern in the forward pass, we add one extra role of a **dQ** writer, since we need to accumulate the value of **dQ** produced by each thread block to the global value of **dQ**. This **dQ** accumulation introduces memory contention (many thread blocks writing to the same location) so having a separate warp to handle this (along with asynchrony) will avoid blocking the rest of the warps in the thread block to perform the next computation (matmul).

We include the backward pass with warp specialization in Algorithm 3.

---

**Algorithm 3** FLASHATTENTION-3 backward pass with warp specialization

---

**Require:** Matrices $\mathbf{Q},\mathbf{K},\mathbf{V},\mathbf{O},\mathbf{dO} \in \mathbb{R}^{N \times d}$ in HBM, logsumexp vector $L \in \mathbb{R}^N$ in HBM, block sizes $B_c, B_r$.

1: In a preprocessing kernel, compute $D = \mathrm{rowsum}(\mathbf{dO} \circ \mathbf{O}) \in \mathbb{R}^d$ (pointwise multiply), write $D$ to HBM and divide it into $T_r$ blocks $D_1,...,D_{T_r}$ of size $B_r$ each.

2: Divide $\mathbf{Q}$ into $T_r = \left\lceil \frac{N}{B_r} \right\rceil$ blocks $\mathbf{Q}_1,...,\mathbf{Q}_{T_r}$ of size $B_r \times d$ each, and divide $\mathbf{K},\mathbf{V}$ in to $T_c = \left\lceil \frac{N}{B_c} \right\rceil$ blocks $\mathbf{K}_1,...,\mathbf{K}_{T_c}$ and $\mathbf{V}_1,...,\mathbf{V}_{T_c}$, of size $B_c \times d$ each.

3: Divide $\mathbf{dO}$ into $T_r$ blocks $\mathbf{dO}_i,...,\mathbf{dO}_{T_r}$ of size $B_r \times d$ each, and divide $L$ into $T_r$ blocks $L_i,...,L_{T_r}$ of size $B_r$ each.

4: Initialize pipeline object to manage barrier synchronization with $s$-stage circular SMEM buffer.

5: **if** in producer warpgroup **then**

6:     Deallocate predetermined number of registers.

7:     Issue load $\mathbf{K}_j$ and $\mathbf{V}_j$ from HBM to shared memory.

8:     Upon completion, commit to notify consumer of the load of $\mathbf{K}_j$ and $\mathbf{V}_j$.

9:     **for** $1 \leq i \leq T_r$ **do**

10:         Wait for the $(i\%s)$th stage of the buffer to be consumed.

11:         Issue loads of $\mathbf{Q}_i,\mathbf{dO}_i$ from HBM to shared memory at the $(i\%s)$th stage of the buffer.

12:         Upon completion, commit to notify consumers of the loads of $\mathbf{Q}_i,\mathbf{dO}_i$.

13:     **end for**

14: **else if** in consumer warpgroups **then**

15:     Reallocate predetermined number of registers as function of number of consumer warps.

16:     On-chip, Initialize $\mathbf{dK}_j = (0)_{B_c \times d}, \mathbf{dV}_j = (0)_{B_c \times d}$ .

17:     Wait for $\mathbf{K}_j$ and $\mathbf{V}_j$ to be loaded in shared memory.

18:     **for** $1 \leq i \leq T_r$ **do**

19:         Wait for $\mathbf{Q}_i$ to be loaded in shared memory.

20:         Load $L_i,D_i$ from HBM to on-chip SRAM.

21:         On chip, compute $\mathbf{S}_i^{(j)} = \mathbf{Q}_i \mathbf{K}_j^T \in \mathbb{R}^{B_r \times B_c}$ (SS-GEMM). Commit.

22:         Wait for $\mathbf{dO}_i$ to be loaded in shared memory.

23:         On chip, compute $\mathbf{dP}_i^{(j)} = \mathbf{dO}_i \mathbf{V}_j^\top \in \mathbb{R}^{B_r \times B_c}$ (SS-GEMM). Commit.

24:         On chip, wait for $\mathbf{S}_i^{(j)}$, then compute $\mathbf{P}_i^{(j)} = \exp(\mathbf{S}_{ij} - L_i) \in \mathbb{R}^{B_r \times B_c}$ .

25:         On chip, wait for $\mathbf{dP}_i^{(j)}$, then compute $\mathbf{dS}_i^{(j)} = \mathbf{P}_i^{(j)} \circ (\mathbf{dP}_i^{(j)} - D_i) \in \mathbb{R}^{B_r \times B_c}$ .

26:         On chip, compute $\mathbf{dV}_j \leftarrow \mathbf{dV}_j + (\mathbf{P}_i^{(j)})^\top \mathbf{dO}_i \in \mathbb{R}^{B_c \times d}$ (RS-GEMM). Commit.

27:         On chip, compute $\mathbf{dK}_j \leftarrow \mathbf{dK}_j + \mathbf{dS}_i^{(j)^\top} \mathbf{Q}_i \in \mathbb{R}^{B_c \times d}$ (RS-GEMM). Commit and wait for both $\mathbf{dV}_j$ and $\mathbf{dK}_j$.

28:         On chip, compute $\mathbf{dQ}_i^{(\mathrm{local})} = \mathbf{dS}_i^{(j)} \mathbf{K}_j \in \mathbb{R}^{B_r \times d}$ (SS-GEMM), and write $\mathbf{dQ}_i^{(\mathrm{local})}$ to smem. Notify the **dQ**-writer.

29:     **end for**

30: **else if** in **dQ**-writer warp **then**

31:     **for** $1 \leq i \leq T_r$ **do**

32:         Wait for $\mathbf{dQ}_i^{(\mathrm{local})}$ to be ready in smem.

33:         Using a semaphore, atomically add $\mathbf{dQ}_i^{(\mathrm{local})}$ to $\mathbf{dQ}_i$ in global memory.

34:     **end for**

35: **end if**

---

## B.2   2-Stage Pipelining SASS Analysis

We give simplified SASS code for the inside of the consumer warpgroup mainloop.

```
// Compute row_max
FMNMX.FTZ R0, R24, R6, !PT ;
SHFL.BFLY PT, R185, R2, 0x2, 0x1f ;
... FMNMX and SHFL.BFLY ...

// Apply exp2 and row_sum. Rescale O.
FMUL.FTZ R2, R4, UR9 ;
MUFU.EX2 R185, R184 ;
FFMA.FTZ R24, R24, UR9, -R6.reuse ;
FADD.FTZ R24, R211, R24 ;
... FMUL, FFMA, FMUL, MUFU.EX2, FADD ...

// FP32 -> FP16 conversion are interleaved with exp2, row_sum and O rescaling.
F2FP.F16.F32.PACK_AB R231, R25, R231 ;
... F2FP, FMUL, MUFU, FFMA, FADD ...

// Start the first WGMMA. Broken down into 8 HGMMAs.
// The first 7 HGMMAs are packed together.
WARPGROUP.ARRIVE ;
HGMMA.64x192x16.F32 R24, gdesc[UR44], RZ, !UPT ;
... HGMMA x 6 ...

// FP32->FP16, exp2, row_sum, O rescaling are interleaved with HGMMA.
F2FP.F16.F32.PACK_AB R214, R214, R187 ;
MUFU.EX2 R234, R5 ;
FADD.FTZ R237, R187, R2 ;
... F2FP, MUFU, FADD ...

// The last HGMMA is issued here. No need to wait.
HGMMA.64x192x16.F32 R24, gdesc[UR44], R24, gsb0 ;

// Start the second WGMMA. Broken down into 12 HGMMAs.
// All 12 HGMMAs are packed together. Not interleaved with other instructions.
WARPGROUP.ARRIVE ;
HGMMA.64x128x16.F32 R120, R228, gdesc[UR8].tnspB, R120 ;
... HGMMA x 10 ...
HGMMA.64x128x16.F32 R120, R184, gdesc[UR8].tnspB, R120, gsb0 ;

// wgmma.wait_group at the end.
WARPGROUP.DEPBAR.LE gsb0, 0x0 ;
```

We make the following observations:

1. Softmax is reordered to the very beginning, even before the first WGMMA.

2. The first WGMMA is interleaved with softmax and FP32 $\rightarrow$ FP16 datatype conversion of **S**. This indicates that WGMMA and non-WGMMAs are executed in parallel.

3. exp2, row\_sum, O rescaling and FP32 $\rightarrow$ FP16 conversions are interleaved together.

4. The second WGMMA is not overlapped with other instructions, as expected.

Overall, SASS shows that the 2-stage pipelining idea works as expected.

## B.3   3-Stage Pipelining Algorithm

We experiment with a 3-stage pipelining algorithm to parallelize the first WGMMA from iteration $j+2$, softmax from iteration $j+1$, and the second WGMMA from iteration $j$. We describe this algorithm in Algorithm 4. This algorithm behaves worse than the 2-stage pipelining algorithm due to the reasons below:

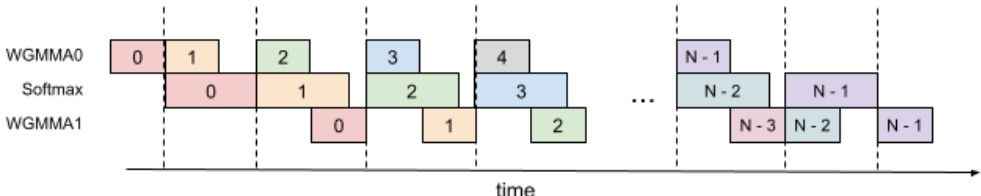

Figure 8: 3-Stage Pipelining

---

**Algorithm 4** FLASHATTENTION 3-stage pipelining consumer warpgroup forward pass

---

**Require:** Matrices $\mathbf{Q},\mathbf{K},\mathbf{V} \in \mathbb{R}^{N \times d}$ in HBM, block sizes $B_c$, $B_r$. Each warpgroup reads 1 block Qi of size $B_r \times d$, $T_c = \left\lceil \frac{N}{B_c} \right\rceil$ blocks $\mathbf{K}_1,...,\mathbf{K}_{T_c}$ and $\mathbf{V}_1,...,\mathbf{V}_{T_c}$ of size $B_c \times d$. Each warpgroup writes 1 output block $\mathbf{O}_i$ of size $B_r \times d$, and 1 logsumexp block $L_i$ of size $B_r$.
1:  Initialization. Load $\mathbf{Q}_i$ from HBM to on-chip SRAM. Initialize $\mathbf{O}_i, \ell_i, m_i, scale\_o$.
2:  Wait for the producer warpgroup loading $\mathbf{K}_0$ from HBM to on-chip SRAM.
3:  Compute $\mathbf{S} = \mathbf{Q}_i\mathbf{K}_0^T$ using WGMMA. Commit and wait.
4:  Compute $m_i, \tilde{\mathbf{P}}_i, \ell_i, scale\_o$ based on $\mathbf{S}$.
5:  Wait for the producer warpgroup loading $\mathbf{K}_1$ from HBM to on-chip SRAM.
6:  Compute $\mathbf{S} = \mathbf{Q}_i\mathbf{K}_1^T$ using WGMMA. Commit and wait.
7:  **for** $2 \leq j < T_c - 2$ **do**
8:      Wait for the producer warpgroup loading $\mathbf{K}_j$ from HBM to on-chip SRAM.
9:      Compute $\mathbf{S}\_next = \mathbf{Q}_i\mathbf{K}_j^T$ using WGMMA. Commit but do not wait.
10:     Wait for the producer warpgroup loading $\mathbf{V}_{j-2}$ from HBM to on-chip SRAM.
11:     Rescale $\mathbf{O}_i$ based on $scale\_o$.
12:     Compute $\mathbf{O}_i = \mathbf{O}_i + \tilde{\mathbf{P}}_i\mathbf{V}_{j-2}$ using WGMMA. Commit but do not wait.
13:     Compute $m_i, \tilde{\mathbf{P}}_i\_next, \ell_i, scale\_o$ based on $\mathbf{S}$.
14:     Wait for all previous WGMMAs.
15:     Copy $\mathbf{S}\_next$ to $\mathbf{S}$.
16:     Copy $\tilde{\mathbf{P}}_i\_next$ to $\tilde{\mathbf{P}}_i$.
17: **end for**
18: Wait for the producer warpgroup loading $\mathbf{V}_{T_c-2}$ from HBM to on-chip SRAM.
19: Rescale $\mathbf{O}_i$ based on $scale\_o$.
20: Compute $\mathbf{O}_i = \mathbf{O}_i + \tilde{\mathbf{P}}_i\mathbf{V}_{T_c-2}$ using WGMMA. Commit and wait.
21: Compute $m_i, \tilde{\mathbf{P}}_i, \ell_i, scale\_o$ based on $\mathbf{S}$.
22: Wait for the producer warpgroup loading $\mathbf{V}_{T_c-1}$ from HBM to on-chip SRAM.
23: Rescale $\mathbf{O}_i$ based on $scale\_o$.
24: Compute $\mathbf{O}_i = \mathbf{O}_i + \tilde{\mathbf{P}}_i\mathbf{V}_{T_c-1}$ using WGMMA. Commit and wait.
25: Epilogue. Rescale $\mathbf{O}_i$ based on $\ell_i$. Compute $L_i$ based on $\ell_i$ and $m_i$. Write $\mathbf{O}_i$ and $L_i$ to HBM as the $i$-th block of $\mathbf{O}$ and $L$.

---

**Overlapping.** We expected that softmax can be overlapped with (the first WGMMA + the second WGMMA). However, the compiler doesn't cooperate in this way. SASS code shows that only the first WGMMA is overlapped with softmax, while the second WGMMA is not. It's not clear why the compiler chooses to reorder instructions in this way.

**Register pressure.** This algorithm requires more registers compared to the 2-stage pipelining algorithm. In theory, it needs to store an extra $\tilde{\mathbf{P}}_i$ and $scale\_o$, which is of size $B_r \times B_c \times$ sizeof(input_data_type) $+ B_r \times$ sizeof(float). As a result, a smaller block size needs to be chosen.

## B.4 Variable Sequence Length

Some optimizations mentioned above cannot be directly used for variable sequence lengths and require special handling.

**TMA** To enable TMA to handle variable sequence lengths directly, additional steps are required. These include modifying a tensormap using the PTX instruction 'tensormap.replace' and store the

tensormap in global memory, which adds overhead and complexity. To address this, during the forward pass when loading Q, we make TMA consistently loads tile_size rows of data. For out-of-bound access, TMA sets zeros for rows beyond the original tensor, while S tensor masking masks out unused rows in a tile. When writing O, we leverage memory-coalesced writes directly, as this is the final step and does not require asynchrony. In the backward pass, a preprocess kernel pads each sequence in dQ, dPSum, and LSE tensors with an additional 128 (tile_size) elements, allowing us to utilize TMA store for efficient data transfer.

**Threadblock cluster and TMA multi-cast**   We utilize TMA multi-cast with a cluster size of 2 for fixed sequence length data loads, allowing every 2 threadblocks processing the same sequence to collaboratively read KV tensors. However, this approach is not well-suited for variable sequence lengths or cases like causal masking and window attention, where some threadblocks may exit earlier and cannot collaborate with other threadblocks in the same cluster. Not utilizing clustering for variable sequence lengths results in a performance drop of around 2% compared to fixed sequence lengths.

### B.5   Masks: causal, local attention, variable sequence length

We apply masks to the S tensor to handle causal and local attention, as well as out-of-bound access for variable sequence lengths. Since masking is expensive, we apply it only when necessary. For instance, in the forward pass, the minimum and maximum KV block indices are calculated and iterated over in the main loop. For causal or variable sequence lengths, masking is applied only to the maximum K block index. For local attention, masking is applied only to the first and last few K block indices based on local attention configurations. Masking is skipped for other K block indices.

### B.6   Persistent Kernel

During the execution of the attention kernel, there is a prologue (loading $Q$) and epilogue (writing output) where the Tensor Cores are not running. To maximize efficiency, we implement a persistent kernel that can overlap the epilogue of one iteration with the prologue of the next iteration to reduce this overhead and keep the Tensor Cores busy. In particular, we launch as many thread blocks as there are streaming multiprocessors (e.g., 132 on the H100 SXM5) and implement a scheduler that assigns tiles to each of the thread block. Each thread block might perform attention for more than one tile.

### B.7   Register data exchange required for second WGMMA in FP8 FLASHATTENTION-3

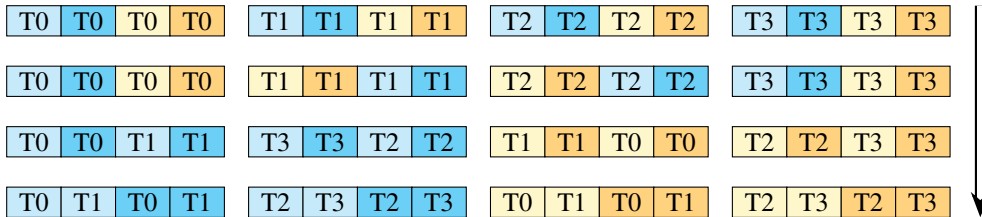

Figure 9: Register data movement to satisfy layout conformance requirements of FP8 WGMMA.

In code, we can effect the register-to-register data exchange that transforms the register ownership pattern of Fig. 3 into Fig. 4 through invoking a combination of the following two CUDA intrinsics:

- `byte_perm`: Given two 32-bit unsigned integers x and y and selector s, the **byte permute** instruction returns 4 bytes from the 8 input bytes as specified by s.
- `shfl_sync`: The **shuffle** instruction exchanges register data from a source lane index j into its own destination register.

Our method is illustrated in Fig. 9. First, we can swap the order of data held within a thread's registers by using byte permute as follows. Referring to the top row of Fig. 9, for a given thread let `upper` be the first 4 bytes (those in light and dark blue) and let `lower` be the last 4 bytes (those in light and dark yellow). Then for the data held by threads 1 and 2, we do the swap by calling `byte_perm` with the indicated selectors:

```
auto upper_mid = __byte_perm(upper, lower, 0x7654);
auto lower_mid = __byte_perm(upper, lower, 0x3210);
```

Now between the second and third rows, we exchange data among threads by using shuffle instructions. Observe that the upper and lower blocks of 4 bytes should be each exchanged among themselves. Moreover, the shuffling of the upper blocks differs from that of the lower blocks, and both shuffles depend on the thread index (mod 4). We account for this using two pre-defined arrays to call `__shfl_sync` with the correct `srcLane` parameter as follows:

```
int upper_map[4] = {0,3,1,2};
int lower_map[4] = {1,2,0,3};
upper_mid = __shfl_sync(uint32_t(-1), upper_mid, upper_map[threadIdx.x%4], 4);
lower_mid = __shfl_sync(uint32_t(-1), lower_mid, lower_map[threadIdx.x%4], 4);
```

Finally, between the third and fourth rows, we repeat the technique with `byte_perm`, but now for all four threads and with the selector depending on the thread index (mod 4). For threads 0 and 3, we have:

```
upper_last = __byte_perm(upper_mid, lower_mid, 0x5410);
lower_last = __byte_perm(upper_mid, lower_mid, 0x7632);
```

whereas for threads 1 and 2, we have:

```
upper_last = __byte_perm(upper_mid, lower_mid, 0x1054);
lower_last = __byte_perm(upper_mid, lower_mid, 0x3276);
```

## B.8  In-kernel transposition of V for FP8 FLASHATTENTION-3

We describe how to fuse the memory transpose of **V** needed for the second FP8 WGMMA into FLASHATTENTION-3. This is handled as an out-of-place SMEM to RMEM to SMEM transfer that is executed in the producer warpgroup.

Specifically, within the producer mainloop, after issuing the TMA load of a tile of **V**, the producer warpgroup waits for the load to complete. Then, producer warps effect the transpose by issuing LDSM (`ldmatrix`) and STSM (`stmatrix`) instructions, which involve a warp of threads collectively loading SMEM to RMEM and storing RMEM to SMEM at a granularity of 128 bytes. Finally, we have an additional pipeline object to manage synchronization between the producer warpgroup and consumers, since the producer pipeline for the TMA load of V now instead has the producer warpgroup as *its* consumer.

We choose LDSM/STSM instructions as they are both register efficient, allowing us to execute them in the producer warpgroup even after register deallocation, and capable of transposing layouts when doing memory copy. Note that as SMEM requirements are first reduced by the smaller memory footprint of the FP8 datatype, we find that we have enough SMEM for the separate buffer used to store the transpose.

There is a technical obstacle to using LDSM and STSM in the context of FP8 datatype that is worth mentioning. Note that in the PTX documentation, LDSM/STSM are described as copying $8 \times 8$ matrices with 16-bit entries [40, §9.7.13.4.15-16], but we can pack 8-bit entries two at a time to use LDSM/STSM in the context of FP8 precision. However, the transpose versions of LDSM/STSM cannot split packed 8-bit entries, which necessitates certain register movements in between LDSM and STSM to actually perform a tile-wise transpose. The use of byte permute to split and reorder packed 8-bit entries in between LDSM and STSM is depicted in the following code snippet:

```
cute::copy(tiled_copy_ldsm, tXsX, tXrX);
auto data = tXrX.data();
#pragma unroll
for (int n = 0; n < size(tXrX); n += 8) {
  uint32_t *data_32bit = reinterpret_cast<uint32_t *>(&data[n]);
  auto upper = data_32bit[0];
  auto lower = data_32bit[1];
  data_32bit[0] = __byte_perm(upper, lower, 0x6420);
  data_32bit[1] = __byte_perm(upper, lower, 0x7531);
}
cute::copy(tiled_copy_stsm, tXrX, tXsX_out);
```

Since this permutes the eventual rows of the transposed **V** tile, we also need to modify the register movements on the consumer side that transform accumulator to operand **P**. We exploit the mathematical fact that

$$\mathbf{P} \cdot \mathbf{V} = \text{colperm}^{\sigma}(\mathbf{P}) \cdot \text{rowperm}^{\sigma}(\mathbf{V})$$

for $\sigma$ a permutation of the common inner dimension of $\mathbf{P}$ and $\mathbf{V}$. Moreover, for the modified register exchange, we can eliminate the use of warp shuffles, but not byte permute, as each thread will already own all the entries it needs for WGMMA.

### B.9 FLASHATTENTION-3 for inference

For decoding inference, the query sequence length is much shorter than the key/value sequence length, typically on the order of one or a few tokens compared to the thousands stored in the KV cache. In this situation, attention becomes a memory-bound workload, and the relevant metric is not tensor core utilization as measured by FLOPs/s, but loading the KV cache as fast as possible as measured by memory bandwidth. Furthermore, since the FLASHATTENTION-3 algorithm described in §3.1 parallelizes over the query sequence length, it can suffer from a lack of parallelism for decoding.

We make two modifications to FLASHATTENTION-3 to introduce more parallelism for decoding:

1. **Split KV** (or **Flash-Decoding**): We split the attention kernel along the key/value sequence length, with the number of splits determined by a heuristic at launch, and combine the resulting outputs using a separate post-processing reduction kernel. "Splitting" according to a parameter $n$ means that $n$ threadblocks load the same tile of $\mathbf{Q}$ and $n$ different segments of the KV cache, computing $n$ different output tiles $\mathbf{O}_1,...,\mathbf{O}_n$ and lse vectors $\mathbf{lse}_1,...,\mathbf{lse}_n$, which we then use to compute $\mathbf{O}$ in the reduction kernel. We also allow for early exit of threadblocks whose given segment of the KV cache doesn't contribute to the final output, in which case the threadblock writes out $-\infty$ as its **lse**.

   This amounts to essentially the same implementation as described in [18].

2. **GQA packing**: For multi-query attention or grouped-query attention, we can restructure the attention mainloop in order to pack multiple query heads per KV head, where each threadblock now loads its $\mathbf{Q}$ tile across different query heads. When query length is short, this achieves additional parallelism "for free" thanks to the large width of the first operand WGMMA tile, given as 64 per warpgroup. For example, we could have a model architecture with 16 query heads per KV head and a query sequence length of 8, in which case a threadblock can pack all 16 query heads into its $\mathbf{Q}$ tile without any change to Algorithm 2. In practice, this yields up to $N$x speedup over an implementation that doesn't do GQA packing, where $N$ is the GQA ratio.

FLASHATTENTION-3 for inference also features an implementation of PagedAttention [29] that was contributed by Kai Londenberg. Recall that PagedAttention is a memory optimization technique for efficiently storing the KV cache in terms of fixed-size pages. This entails separating the logical position of KV blocks from their physical addresses, with a *block table* defining the address translation [29, §4.2].

Now, prior implementations of TMA load in CUTLASS construct the tensor map object such that TMA tensor coordinates are determined using the physical GMEM tensor. To use a block table with TMA, Londenberg defines a new `SM90_TMA_LOAD_PAGED_OP` class and a tensor map constructor that instead determines TMA tensor coordinates in terms of the virtual shape. The block table is then passed into the TMA copy method as an additional argument.

## C  Addition Details on Experiments and Benchmarking

### C.1  System and libraries

We benchmark the speed on an H100 80GB SXM5 (700W). We generally use the latest versions of the libraries, at the time of writing (October 2024). Specifically, we use:

- CUDA 12.3
- cuDNN 9.5.0.50
- CUTLASS 3.6
- FLASHATTENTION 2.6.3
- Triton 3.1
- PyTorch 2.5.0

To reduce variability, we fix the GPU clock speed to 1830MHz (clock speed used to calculate the 989 TFLOPS FP16 theoretical max throughput). We repeat the benchmarks 10 times and take the average timing.

## C.2 FP8 Attention Full Results

We use following sequence lengths: 512, 1024, 2048, 4096, 8192, 16384.

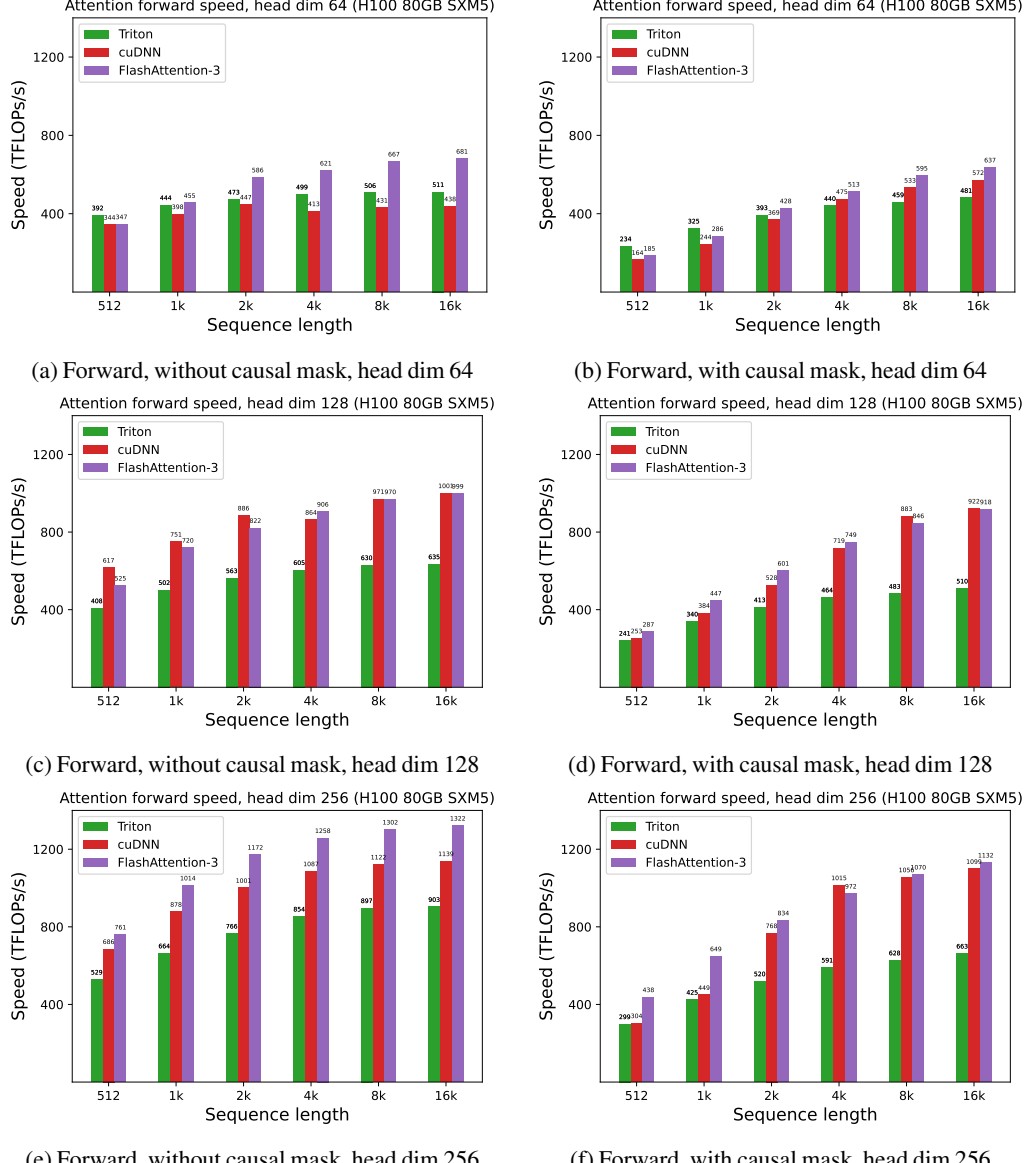

Figure 10: Attention forward speed (FP8) on H100 GPU

