# OpenReview forum: "FlashAttention-3: Fast and Accurate Attention with Asynchrony and Low-precision"
_NeurIPS.cc/2024/Conference — NeurIPS 2024 spotlight_

### Official Review · Reviewer_TcT4 · 2024-07-09

**Soundness:** 3
**Presentation:** 3
**Contribution:** 3
**Rating:** 7
**Confidence:** 3

**Summary:**

FLASH ATTENTION sped up attention on GPUs but achieved only 35% utilization on H100 GPUs. To address this, FLASH ATTENTION-3 introduces three techniques for Hopper GPUs: warp-specialization to overlap computation and data movement by assigning warps to producer and consumer roles; interleaving operations to combine block-wise matrix multiplication (matmul) and softmax operations for asynchronous execution; and support for FP8 precision using block quantization and incoherent processing to leverage hardware capabilities. These techniques result in a 1.5-2.0× speedup with FP16, up to 1.2 PFLOPs/s with FP8, and 2.6× lower numerical error than baseline FP8 attention, improving computational efficiency and hardware utilization on modern GPUs.

**Strengths:**

The strengths of the paper are that it is well presented and shows good results. The paper addresses the important topic of accelerating Transformer modules by presenting an incremental improvement over previous work, specifically FLASHATTENTION-2, and applies these advancements to the new H100 GPU. The paper demonstrates how the enhanced methods leverage the capabilities of the H100 GPU to achieve significant performance gains, showcasing advancements in computational speed and efficiency. This progression highlights the continuous evolution in optimizing Transformer architectures for cutting-edge hardware, emphasizing the relevance and impact of the research in the field of deep learning and hardware acceleration.

**Weaknesses:**

The work primarily focuses on the H100 GPU, demonstrating the performance enhancements of FLASHATTENTION-3 for this hardware. However, evaluating how FLASHATTENTION-3 performs across other GPU families would increase the work's impact, showing its versatility and potential for wider adoption in different computing environments.

**Questions:**

A few question and notes are listed below:

How would the methods are applicable to other GPU devices?

Is speed is slower for flashattention3 in small sequence len only seen in H100?

In 4.1: '4·seqlen2 ·head dimension·number of heads'. would be better to format as equation

**Limitations:**

The authors adequately addressed the limitations and, if applicable, potential negative societal impact of their work

---

> ### Author Rebuttal · Authors · 2024-08-07
>
> We thank the reviewers for the support and appreciate the thoughtful questions.
> 1. Other hardware: please refer to the common response. We are collaborating with other researchers and engineers on FA3 for AMD cards, Google TPUs, and Nvidia Ada cards (e.g. 4090). As suggested by the reviewer, this would enable wider adoption of FA3.
> 2. Small seqlens: FA3 in May was slower than cuDNN for small seqlens (e.g. 512, 1024). We have since developed a persistent scheduler that loads the Q, K, V for the next subproblem while writing out the output of the current subproblem, and thus matching / exceeding cuDNN for small seqlens (FA3 was already faster than cuDNN for large seqlens). This overlapping is important since the prologue / epilogue (loading initial Q, K, V and writing out O) takes non-negligible time (e.g. up to 10-15% of the overall computation) when the sequence is short.

---

> > ### Comment · Reviewer_TcT4 · 2024-08-07
> > **rebuttal acknowledgment**
> >
> > thank you for clarifications

---

### Official Review · Reviewer_KCED · 2024-07-12

**Soundness:** 3
**Presentation:** 3
**Contribution:** 4
**Rating:** 8
**Confidence:** 4

**Summary:**

The paper builds on the existing work on FlashAttention, and introduces an advanced method to accelerate the attention mechanism in Transformer model. It specifically targets the newer GPU architectures, like NVIDIA H100, by exploiting asynchrony in Tensor Cores and Tensor Memory Accelerators. There are three techniques that are proposed: 1) producer-consumer asynchrony across warps; 2) compute asynchrony within consumer warps; 3) hardware accelerated low precision GEMM. The extensive empirical results show impressive performance, with significant speedups compared to state-of-the-art methods, and with reduced numerical errors.

**Strengths:**

The authors present novel techniques that exploit hardware features like asynchrony and low-precision computation.
The performance gains are impressive, in speed and accuracy.
The empirical validation includes comprehensive benchmarks and comparisons with previous methods.
It is nice to see the open source commitment.

**Weaknesses:**

The proposed methods are highly complex, and may be difficult to implement and understand for practitioners.
The techniques are tailored to NVIDIA Hopper GPUs, and may have limited applicability to other hardware achitectures.

**Questions:**

The backward pass algorithm is only included in the appendix. It would be useful if it could be included in the main paper.
Can you comment if FlashAttention-3 could be used on older GPU architectures that lack some of the advanced features of the Hopper GPUs?

**Limitations:**

Yes.

---

> ### Author Rebuttal · Authors · 2024-08-07
>
> We are very happy that the reviewer appreciates the impact of the paper.
> 1. Generality: we are excited about generalizing our techniques for other hardwares (AMD GPUs, Google TPUs, and Nvidia Ada GPUs). Please refer to the common response for more details.
> For Ampere cards (e.g. A100, 3090) and Ada cards (e.g. 4090), with BF16/FP16, FA2 is already close to optimal (reaching 70% of theoretical max FLOPS, while matmul can reach around 70-80% of theoretical max FLOPS). For Ada cards that support FP8 (e.g. 4090), there is an opportunity to potentially double the attention throughput. Though they lack asynchronous Tensor Memory Accelerator (TMA) and the matmul instructions are synchronous (mma.sync instead of wgmma.mma_async), there’s still some form of asynchronous copy (cp.async). Though it might be harder to overlap GEMM and softmax, we are still exploring FP8 on Ada cards, since they offer the highest FLOPS per dollar.
>
> 2. Backward pass: it is more involved than the forward pass, and we would include it in the main text if space permits.

---

### Official Review · Reviewer_qxbQ · 2024-07-12

**Soundness:** 3
**Presentation:** 3
**Contribution:** 4
**Rating:** 7
**Confidence:** 5

**Summary:**

This is a system paper that introduces FlashAttention-3, an optimized version of FlashAttention for NVIDIA's SM90+ GPUs. The key contributions (summarized by the authors) are:

- Taking advantage of warp specialization in NVIDIA SM90+ and designing a producer-consumer computation paradigm to allow better intra-CTA overlapping between producers and consumers;
- Designing 2-stage software pipeline to allow better overlapping within consumers;
- Designing on-the-fly layout transformation to support FP8 WGMMA and using incoherent processing to reduce quantization error by 3x.

**Strengths:**

- The empirical results of this paper are very strong, and are expected to generate immediate impact to the community after code release (promised by the authors).
- The presentation of this paper is clear. It is easy for people who are familiar with NVIDIA SM8x but not experts in SM90+ to follow the writing and essence of the optimizations. I learned a lot from the descriptions related to the warp specialization design.
- This is the first (to-be) open-sourced system paper that discusses the challenges and solutions related to FP8 attention, which has been implemented and close-sourced by NVIDIA for a long time. The insights related to FP8 such as layout transformation and reducing quantization error are very helpful for the readers.

**Weaknesses:**

[Results, Benchmarking]

- There has already been a paper from Colfax (arXiv 2312.11918) available online more than half a year ago describing how to implement efficient FlashAttention on Hopper GPUs. It will be better if the authors could cite this paper and compare the performance of FlashAttention-3 and this paper.
- NVIDIA's TensorRT-LLM has a close-sourced implementation of FP8 attention. The result comparison with this library is missing.
- While the authors mention RMSE reduction from incoherence processing using simulated data, there is no evaluation on how this step improves real LLM inference workloads (such as Wikitext perplexity, MMLU accuracy, etc.)
- How many end-to-end LLM inference / training speedup can we obtain by switching to this FP8 attention? For inference, I am interested in a TensorRT-LLM-like implementation with FP8 GEMM + FP16 attention (FlashAttention-3) VS FP8 GEMM + FP16 attention (FlashAttention-2) VS FP8 GEMM + FP8 FlashAttention-3.

[Methodology, Novelty]

- While all the methodologies mentioned in the paper make sense to me, it is important to know that some of them occurred in previous literature / open source implementation on the same workload. For example, FlashInfer from University of Washington implements within-consumer software pipelining on SM80+ devices. Incoherent processing is also not new, and has been adopted in related work such as QUIP#, QuaRot, etc.
- The authors mentioned that a lot of operations can be fused into the preceding RoPE operation **with no additional overhead**. For example, transposing V, applying Hadamard transformation, block quantization, etc. While I agree that intuitively computation operations could be fused with memory bound Ops, it is also important to notice that the CUDA cores of H100 are not that strong. I want to see real measured numbers in end-to-end workloads that justify the authors' claims on "no additional overhead".

**Questions:**

Please address my comments in the Weaknesses section. Additional questions related to ablation studies:

- Will it be also possible to open source the benchmarking code in Table 2? This will be super helpful for system researchers to understand the impact of warp specialization and 2-stage computation pipeline on the end-to-end throughput in different workloads.
- Please comment why your implementation achieves performance advantage over OpenAI Triton ones. What is the most important design space extension that leads to this improvement?
- Please comment on how the register re-allocation feature provided by NVIDIA SM90+ impacts the performance. Are there any important design spaces enabled by this feature which would otherwise lead to register spilling to local memory in SM8x?
- Please comment on whether GEMM-SoftMax 2-stage pipelining bring about performance improvement on SM8x (it's related to the question above. If there are a lot of register spills then I think this is also a specialized optimization for SM90+).

**Limitations:**

The authors have adequately addressed the limitations of this paper. There is no potential negative societal impact.

---

> ### Author Rebuttal · Authors · 2024-08-07
>
> Thank you for the thorough review and helpful suggestions.
>
> 1. Existing literature: The Colfax arXiv paper discusses use of WGMMA and TMA for FA2, but not more sophisticated techniques with asynchrony. This is similar to the Triton implementation that we compared to. We will cite and include this in the discussion. FA3, with asynchrony (warp specialization, GEMM-softmax overlapping) can reach 800 TFLOPS for headdim 256 FP16, while the Colfax implementation can reach 550 TFLOPS for the same setting. This highlights the importance of the new algorithm.
>
> We have benchmarked the closed-source TensorRT-LLM implementation and found its speed similar to the cuDNN implementation. We have included the comparison with cuDNN in the paper. In the newest version, FA3 is about 10-25% faster than cuDNN for FP16/BF16, even though cuDNN was already optimized for Hopper GPUs. We are hopeful that by open-sourcing FA3, other researchers can build on it to find other hardware-aware algorithms such as approximate (sparse, low-rank) attention.
>
> We plan to also open source the benchmarking code, thanks for this suggestion.
>
> 2. End-to-end evaluation: we include this in the common response, in an LLM inference production setting.
> We include a more detailed experiment here, measuring the savings from switching FA2 -> FA3 on Llama-3.1 405B with different context length (percentage of time saved vs original time). Note that GEMMs are done in FP8 and attention is done in BF16.
>
> | Model size / Seq length | 8K | 16K | 32K | 128K |
> |-------------------------|-----|------|------|-------|
> | 405B | 4.6%| 7.3% | 5.4% | 26.2% |
>
> We see that even for some of the largest models (where MLPs and QKV projections are very large), we can save up to 26.2% time, i.e. 1.35x speedup. Roughly speaking, for 405B at 128K sequence length, attention takes 50% of the time and MLP takes 50% of the time, so speeding up attention by 2x would bring around 1.3x end-to-end speedup.
>
> 3. Existing methodologies: we leverage existing techniques and build on a rich literature. We will add these to the discussion. We find it very encouraging that these conceptually simple techniques can speed up attention, one of the core operations in ML and has been studied intensely by the community, by up to 2x on modern hardware.
>
> 4. Fusing operations: we show here the time taken by RoPE, Hadamard transform, and their fusion.
> We measure on an H100 80GB SXM5 (3.35TB/s memory bandwidth), with Q and K of size 2 x 8192 x 32 x 128 each (BF16).
> | Operation | Timing |
> |-----------|-----|
> | Memcopy | 110us|
> | RoPE | 115us |
> | Hadamard transform | 107us |
> | RoPE + Hadamard transform | 120us |
> We see that all of the combinations reach about 70-80% of memory bandwidth.
> 5. Comparison with Triton: The most significant improvement of FA3 over it is fine-grained exploitation of asynchrony / scheduling techniques, Though Triton already uses Hopper-optimized instructions (WGMMA and TMA), there is no automatic warp-specialization or overlapping between GEMMs and other operations like softmax yet. We are collaborating with Triton developers to implement some of these techniques in the compiler.
> 6. Register re-allocation is important for warp-specialization, since the warps doing copying (TMA) needs very few registers while the warps doing matmul (WGMMA) need a lot of registers to hold the accumulator, and more registers means we can use larger WGMMA instructions and more opportunity to overlap with other operations. The other techniques (e.g. overlapping GEMM and Softmax) can still apply without warp-specialization / register re-allocation.
> 7. Ampere / Ada GPUs (SM8x). One can still overlap GEMM and softmax in SM8x, but warp specialization is difficult and generally not done (e.g. for matmul). We note that FA2 is already close to optimal on Ampere / Ada GPUs, reaching up to 70% of theoretical max FLOPS (while matmul can reach around 80%). For newer hardware, the tensor cores are simply much faster, so there is a greater need for asynchrony / overlapping. We expect this trend to hold for future accelerators (please see the common response for more detailed discussion). We are also exploring FP8 on Ada, even without warp specialization we believe that FP8 can still substantially improve the attention throughput on Ada cards.

---

> > ### Comment · Reviewer_qxbQ · 2024-08-14
> >
> > Thanks. The rebuttal has addressed my concerns.

---

### Official Review · Reviewer_WEiA · 2024-07-13

**Soundness:** 4
**Presentation:** 4
**Contribution:** 3
**Rating:** 8
**Confidence:** 4

**Summary:**

This paper presents FlashAttention-3, which speedup the commonly-used attention operator on Hopper GPUs. The paper proposes to leverage the asynchronous execution of the Tensor Cores and Tensor Memory Acceleator to better utilize the GPU hardware. Specifically, the paper proposes three techniques: (1) It overlaps the data movement and GEMM computation by warp specialization; (2) It proposes to overlap the GEMM and softmax operations by interleaving these operations; and (3) It leverages FP8 tensor core by block quantization. The experimental results show that FlashAttention-3 can achieve superior speedup by 1.5-2.0x with FP16 and can outperform vendor libraries.

**Strengths:**

* It optimze attention operator on Hopper architecture, which is a timely problem and is of great importance;
* It provides new insights for optimizing for Hopper architecture;
* It achieves superior speedup compared to prior work and comparable speedup to vendor libraries;

**Weaknesses:**

* The method introduces a few hyper-parameters such as number of pipeline stages, block sizes $B_r$ and $B_c$. However, the paper has never discussed how to tune or select this parameters in practice.
* The paper does not evaluate with the end-to-end LLM inference task, which makes it obscure to see its real performance gain in practice.

**Questions:**

Thanks for submitting the excellent paper to NeurIPS. The paper is in general well-written and the ideas are novel and insightful. However, I would like to make the following comments for further polishing the paper:

The paper has mentioned a few hyperparameters. However, how do you tune these hyperparameters to achieve the best performance? Do you use grid search or simply heuristics? This is important because the inputs to language models often have varying sequence lengths, and tuning each of them is not practical. It would be great to see a systematic way to find these hyperparameters.

The paper achieves superior speedup on the single flash attention operator. However, given the existence of FlashAttention-2, how much percentage does the attention operation takes in the context of whole LLM inference or training pipeline? This could help better understanding the positioning of this paper.

**Limitations:**

* The paper is only specifically optimized for the Hopper architecture and may not applies to other or future GPUs.

---

> ### Author Rebuttal · Authors · 2024-08-07
>
> We appreciate the enthusiastic support from the reviewer.
>
> 1. Setting of hyperparameters: We select tile size and stage count hyperparameters as a function of the head dimension (64, 128, 256) and datatype (16 or 8 bit) based on available register and smem budget. This is similar to matrix multiply (e.g. from cuBLAS or Triton), where the tile sizes and number of stages are tuned either with heuristics (cuBLAS) or by an autotuner (Triton). For FA3, we found 2 stages to be consistently best, and we choose tile sizes (B_r and B_c) to simply maximize the registers and shared memory that we can use. There are not that many choices for these tile sizes, (e.g. 128 x 128, 128 x 256, 256 x 128) since some dimensions have to be divisible by 64 or 128 due to hardware constraint, so we tune them by hand. These hyperparameters are not functions of seqlen, so different sequence lengths use the same tile sizes.
> 2. For LLM inference, FA3 would be most helpful for the prefill stage. For the decode stage, the bottlenecks are different (loading KV cache as fast as possible), and other techniques are more relevant (e.g Flash-Decoding, KV cache quantization).
>
> Thanks for the suggestion on end-to-end evaluation. We have mentioned the impact of FA3 on LLM inference in a production setting in the shared response. We include more detailed experiment here, measuring the savings from switching FA2 -> FA3 on Llama-3.1 405B with different context length (percentage of time saved vs original time).
>
> | Model size / Seq length | 8K | 16K | 32K | 128K |
> |-------------------------|-----|------|------|-------|
> | 405B | 4.6%| 7.3% | 5.4% | 26.2% |
>
> We see that even for some of the largest models (where MLPs and QKV projections are very large), we can save up to 26.2% time, i.e. 1.35x speedup. Roughly speaking, for 405B at 128K sequence length, attention takes 50% of the time and MLP takes 50% of the time, so speeding up attention by 2x would bring around 1.3x end-to-end speedup.
> At smaller model sizes, the speedup is larger at the same sequence length, since attention would take proportionally more time.

---

> > ### Comment · Reviewer_WEiA · 2024-08-08
> > **Rebuttal Acknowledgment**
> >
> > Thanks for the clarifications. I have read the rebuttals.

---

### Author Rebuttal · Authors · 2024-08-07

We thank the reviewers for their enthusiastic support, their careful read of the paper, and their thoughtful questions and suggestions. We are very happy that the reviewers find the paper “has immediate impact to the community”, the ideas “novel and insightful”, and the writing “clear and easy to understand”.

Since the submission in May, we have made FA3 even faster, more accurate, easier to use, and more general:
- New optimizations: (1) A persistent scheduler to start loading the Q, K, V for the next subproblem while writing out the output O of the current subproblem, thus speeding up the cases of small seqlen (2) Optimized for the case where sequences in the batch have variable lengths (e.g. in Llama 3 training) (3) Reduced the numerical error for GQA/MQA by accumulating the gradient dK and dV of different heads in FP32, before converting the sum to BF16. This brings 2-3x smaller numerical error compared to the standard method of computing dK and dV for different heads in BF16, then summing the gradients.
- Speed: FA3 version in August is 6-25% faster, reaching up to 700 TFLOPS for headdim 128 FP16 (vs 650 in May), 800 TFLOPS for headdim 256 FP16 (vs 640 in May), and 1300 TFLOPS for headdim 256 FP8 (vs 1230 in May). This is thanks to a technique called pingpong scheduling that better overlaps the GEMM of one warpgroup with the softmax of another warpgroup. For FP16/BF16, attention speed from FA3 is now on par with matmul speed from cuBLAS (arguably the most optimized operation) for similar sizes (700-770 TFLOPS), suggesting that we are using the hardware as efficiently as possible. This is another validation that asynchrony can play a major role in optimizing for modern hardware.
- Integration with other libraries: we are working with PyTorch developers to integrate FA3 to PyTorch to benefit the largest number of researchers and engineers. We are also working with inference and training libraries (Hugging Face, vLLM, Megatron-LM, TransformerEngine, DeepSpeed) to help with their integration effort to speed up transformer training and inference with FA3.
- Generality: We are working with Triton and Jax/XLA developers to implement some of the techniques in FA3 in the Triton & XLA compilers. We are also collaborating with other researchers and engineers on hardware other than H100 (elaborated below).

We now respond to some common questions from the reviewers.
1. **Generality**: The techniques developed in FA3 are not limited to Nvidia Hopper GPUs. As mentioned in the intro, asynchrony and low-precision are the general trend of AI accelerators, due to Amdahl's law. Amdahl’s law states that the overall performance improvement gained by optimizing a single part of a system is limited by the fraction of time that the improved part is actually used. As tensor cores in hardware accelerate matmuls exponentially with each new generation of hardware, the serial sections of the code (softmax and causal masking in attention) can dominate the runtime. The only way to overcome these latencies, other than adding expensive hardware units that compute them, is to overlap their execution with other concurrent work. We chose to validate our algorithms and ideas on Hopper GPUs due to good hardware and software support (async barriers, CUTLASS), but these ideas also apply to other modern accelerators. In fact, we are collaborating with AMD engineers to implement the overlapping of GEMM and Softmax on AMD GPUs such as MI300X, as well as Google researchers on a version of FA3 on TPUs. We are also exploring FP8 support for Ada GPUs (e.g. 4090) to potentially double the attention throughput on these consumer cards.
2. **Impact**: With FA3, we have significantly sped up attention, one of the two main layers in Transformers. Attention speedup can have a major impact when the sequences are long, which is increasingly common (e.g. Llama 3.1 has 128k context length). In a LLM inference production setting (with an already very optimized inference engine), we have measured that switching from FA2 to FA3 yields up to 30-40% speedup in time to first token for Llama-3 405B. This is a major improvement for inference services running on hundreds or thousands of GPUs. For 405B at 128K sequence length, attention takes about 50% of the time and MLP takes 50% of the time, and making attention 2x faster will yield about 1.3x speedup. One can combine FA3 with other techniques to speed up the MLP (e.g. sparsity, FP6/FP4) to get even higher speedup.
3. **Interoperability**: FA3 is a drop-in replacement for FA2, save for the FP8 case where the version of FA3 in May requires V to have layout BHDS (sequence length dimension must be contiguous), due to the constraint of the FP8 tensor cores. We have since developed a variant of the FP8 kernel that supports V input in the standard BSHD layout (with the head dimension as the contiguous mode). This is important to integrate FP8 attention into standard libraries, such as PyTorch and vLLM, and also circumvents a technical challenge with variable sequence length in terms of loading memory addresses not aligned to 16 bytes.
Recall from the paper that the FP8 WGMMA instruction only supports k-major operand B, so this variant necessitates doing an "in-kernel" transpose on tiles of V after they are loaded from GMEM to SMEM. Moreover, in order to overlap this transpose with other operations to minimize its impact on speed, we change the kernel scheduling. In short, we take advantage of special LDSM-STSM instructions that both minimize register usage and support transposition for SMEM <=> RMEM copy in order to place the in-kernel transpose of V in the producer, thereby further leveraging warp specialization with warpgroup register reallocation.
Our implementation achieves comparable performance to cuDNN, exceeding it for headdim 64 and matching for headdim 128/256.

Overall we are excited about generalizing our techniques to other hardwares to unlock new use cases for long context models.

---

### Decision · Program_Chairs · 2024-09-25

**Decision:**

Accept (spotlight)

**Comment:**

The work introduces FlashAttention 3, an optimized version of FlashAttention for modern GPUs, which is a key building block for efficient transformer models. While a main focus is on NVIDIA's Hopper GPUs which currently lack such efficient kernels, the innovations of this paper are also promising and valuable for GPUs from other vendors and various other compute backends. Compatibility with new lower-precision floating point formats such as FP8 is another strength of this work, which is very valuable for the wider community.